# Towards Semi-Structured Automatic ICD Coding via Tree-based Contrastive Learning

**Chang Lu⋆, Chandan K. Reddy†, Ping Wang⋆, Yue Ning⋆**
⋆ Department of Computer Science, Stevens Institute of Technology
† Department of Computer Science, Virginia Tech
⋆ {clu13, ping.wang, yue.ning}@stevens.edu
† reddy@cs.vt.edu

## Abstract

Automatic coding of International Classification of Diseases (ICD) is a multi-label text categorization task that involves extracting disease or procedure codes from clinical notes. Despite the application of state-of-the-art natural language processing (NLP) techniques, there are still challenges including limited availability of data due to privacy constraints and the high variability of clinical notes caused by different writing habits of medical professionals and various pathological features of patients. In this work, we investigate the semi-structured nature of clinical notes and propose an automatic algorithm to segment them into sections. To address the variability issues in existing ICD coding models with limited data, we introduce a contrastive pre-training approach on sections using a soft multi-label similarity metric based on tree edit distance. Additionally, we design a masked section training strategy to enable ICD coding models to locate sections related to ICD codes. Extensive experimental results demonstrate that our proposed training strategies effectively enhance the performance of existing ICD coding methods.

## 1 Introduction

The adoption of electronic health records (EHR) data has become widespread in modern healthcare facilities as they provide a centralized platform to maintain comprehensive medical information of patients, including diagnoses, procedures, laboratory tests, and clinical notes [1]. To efficiently manage and categorize diseases and procedures, EHR data utilizes the International Classification of Diseases (ICD) system developed by the World Health Organization. The ICD system provides a hierarchical structure that maps diseases/procedures to digital codes. Clinical notes in EHR data are generally stored as free text, while diagnosis and procedure codes are extracted from these notes and saved as structured data. The process of extracting ICD codes from clinical notes is referred to as **ICD coding** and is a crucial task in medical services such as medical records management, medical billing [23], and insurance reimbursement [19]. It also supports healthcare research endeavors such as diagnosis prediction [1, 15] and medication recommendation [22].

The traditional ICD coding task relies on human effort, which is both time-consuming and prone to errors [27]. Incorrect code assignments can be costly. For instance, the error payout rate due to wrong code assignment reached 6.8% in 2000, as stated by the US Centers for Medicare and Medicaid's statistics [17]. Consequently, researchers are exploring automated ICD coding methods to assign ICD codes to medical documents with algorithms. Recent methods generally treat the ICD coding task as a multi-label classification problem [13, 32, 30], as one clinical note can contain multiple diagnosis/procedure codes. To capture the relationship between text and codes, code representations have been studied by incorporating the semantic information of code names [18] with hierarchical structures, synonyms, and co-occurrence of codes to provide fine-grained code representations and

37th Conference on Neural Information Processing Systems (NeurIPS 2023).

relationships [4, 30, 29]. However, due to the nature of clinical notes, automatic ICD coding tasks still present *certain challenges* when it comes to learning the representation of clinical notes:

1. **Limited availability of data.** Due to privacy constraints, EHR data can be challenging to access. For example, a publicly available EHR dataset, MIMIC-III [11], only contains around 50,000 clinical notes that can be used for ICD coding. Furthermore, the appearance of codes in an EHR dataset follows a long-tail distribution. Around 51.6% of codes occur less than 6 times, and 60.2% of codes occur less than 10 times, making it even more difficult to train data-driven deep learning models for these codes due to data paucity.

2. **Ignoring structural information.** Based on our observation, most clinical notes have common sections such as "physical exam", "history of present illness", "discharge followups", and "brief hospital course", as depicted in Figure 1(a). These sections reflect correlated health information in long clinical notes. However, most of the existing ICD coding models treat clinical notes as a single sequence, disregarding these semi-structures that represent essential elements of diagnoses.

3. **Variability of clinical notes.** Clinical notes are written by various medical professionals, who may have different writing habits. Different clinical notes may present different orders and combinations of sections, adding to the variability of the data. Additionally, different patients may also lead to a diverse content of clinical notes due to their unique diseases or physical exams. This variability becomes a more negative factor in training an ICD coding model with limited data.

In this paper, our goal is to utilize the semi-structured format and reduce the variability of clinical notes with limited data. As shown in Figure 1(b), existing methods typically treat clinical notes as long sequences of words [16, 30, 29] without considering the semi-structured format of clinical notes. As a result, these models can easily be affected by the variability of clinical notes with limited data. To overcome this challenge, we propose to automatically segment a clinical note into multiple sections to build an order-agnostic structure. Based on the extracted sections, we introduce a contrastive learning framework to initially reduce the variability in clinical notes in pre-training and allow the text encoder to better understand the relationship of sections with limited training data. This proposed contrastive learning method defines a soft multi-label similarity between section pairs from the same and different clinical notes. Finally, we design a masked section training method to further minimize the variability of clinical notes in the training of ICD coding models.

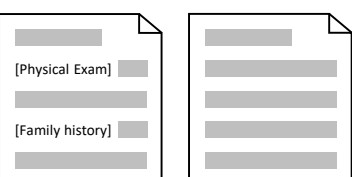

(a) Sections in a clinical note (b) A clinical note as a sequence

Figure 1: An example of a clinical note in the format of multiple sections and one sequence of words.

**Main contributions** In summary, the main contributions of this paper are listed as follows:

- We propose a content-based algorithm that automatically segments clinical notes into sections. To the best of our knowledge, our work is one of the first to investigate automatic semi-structured segmentation for clinical notes in ICD coding.

- We present a contrastive learning framework based on a soft multi-label similarity with tree edit distance and a masked section training strategy to alleviate the variability of clinical notes with limited EHR data.

- We conduct extensive experiments on real-world EHR datasets and demonstrate that our proposed section-based learning can enhance the performance of existing ICD coding methods.

## 2  Related work

The task of ICD coding involves predicting ICD codes from discharge summaries and can be approached as a multi-label text categorization problem. In the past, natural language processing (NLP) techniques have been commonly used to learn the representation of clinical notes. For instance, Perotte *et al.* [21] used the term frequency-inverse document frequency (TF-IDF) features of clinical notes and employed support vector machine (SVM) classifiers for ICD coding. Mullenbach *et al.* [18] proposed CAML based on convolutional neural networks (CNN), while Baumel *et al.* [2] employed a two-layer recurrent neural network (RNN) to encode clinical notes. Liu *et al.* [14] applied squeeze-and-excitation networks in CNN and used the focal loss to deal with rare codes.

In addition to the conventional RNN/CNN-based models, attention and graph neural networks have also been explored in the context of ICD coding. Li *et al.* [13] proposed a multi-filter residual CNN model incorporating label attention between codes and text. Xie *et al.* [28] utilized the hierarchical structure of ICD codes and developed a graph neural network to capture the relation between codes. Cao *et al.* [4] considered both the hierarchical structure and the co-occurrence of ICD codes, embedding the codes into a hyperbolic space. Yuan *et al.* [30] proposed to improve the matching of the ICD code names that occur in clinical notes using attention with code synonyms.

To alleviate the lack of training data and labels, weak supervision has also been studied for ICD coding. It aims to automatically generate weakly labeled training data using rules, heuristics, or medical domain knowledge. Dong *et al.* [7] adopted an existing named entity linking tool called SemEHR to identify rare diseases from clinical notes. Gao *et al.* [8] proposed a labeling function called KeyClass by extracting n-grams as keywords and computing the cosine similarity of word embeddings between keywords and labels.

Although large language models (LLMs) are popular and effective in many NLP tasks such as machine translation and question-answering systems, it has been shown by Pascual *et al.* [20] and Ji *et al.* [10] that pre-trained LLMs such as BERT [6] do not help to improve the performance of ICD coding due to the long text of clinical notes and the difference of training data between ICD coding and pre-training tasks of LLMs. To overcome these problems, Liu *et al.* [9] split the clinical notes into chunks to fit the pre-defined maximum input length of transformer-based models. However, splitting a document into chunks can break coherent information in clinical notes. Yang *et al.* [29] introduced KEPT, a transformer-based model that uses Longformer [3] to encode the long text, pre-trained with a contrastive learning method for code synonyms. They also designed a prompt learning framework for the prediction. However, the Longformer and prompt learning used in KEPT require a huge number of model parameters and extremely long input, which is barely applicable in training.

As previously discussed, most current methods only treat clinical notes as long sequences without considering their semi-structured format, making it challenging to handle the variability of clinical notes. Additionally, while some models, like KEPT, incorporate pre-training, it is designed only for labels but not for clinical notes. Thus, these models are not effective in comprehending the relationship among different sections of clinical notes. In light of these limitations, our paper aims to investigate the semi-structured format of clinical notes and improve the model's ability to learn the representations of long-text clinical notes with limited data.

## 3 Preliminaries

**Problem formulation**  Consider the ICD codes as a set $\mathcal{L} = \{l_i\}_{i=1}^L$, where $L = |\mathcal{L}|$ is the number of codes. Specifically, $l_i = \{w_j\}_{j=1}^m$ with $m$ tokens is the description of the $i$-th label, where $w_j \in \mathcal{V}$, and $\mathcal{V}$ is the vocabulary of all tokens in the ICD code descriptions and clinical notes. Given a clinical note $S = \{w_j\}_{j=1}^n$ with $n$ tokens, the ICD Coding task is to train a model $\mathcal{M}$ to predict a binary vector $\hat{\mathbf{y}} \in \{0, 1\}^L$, where $\hat{\mathbf{y}}_i = 1$ means the code $l_i$ exists in the clinical note $S$.

**General ICD coding framework**  To better demonstrate key parts of the ICD coding, we simplify it as flat multi-label classification. A general ICD Coding framework $\mathcal{M}$ contains three modules:

- *Clinical note encoder* (Enc_note): Given a clinical note $S$, the clinical encoder is a text encoder Enc_text that first encodes the words into embeddings and uses RNN, CNN, or Transformer encoder to compute hidden representations $\mathbf{h}_{\text{note}}$ of words: $\mathbf{h}_{\text{note}} = \text{Enc}_{\text{text}}(S) \in \mathbb{R}^{n \times d}$.

- *ICD code encoder* (Enc_code): This module can be regarded as a domain knowledge encoder that incorporates the text description of all codes (i.e., code names) in the ICD system, which are agnostic to the training-data. It is also a text encoder that first calculates the hidden word representations of a code name and then uses a pooling layer (e.g., mean/max pooling) on words to get the code representation for one ICD code: $\mathbf{h}_{\text{code}}^i = \text{Pooling}(\text{Enc}_{\text{text}}(l_i)) \in \mathbb{R}^d$. Eventually, we have the hidden representations of all the ICD codes: $\mathbf{h}_{\text{code}} \in \mathbb{R}^{L \times d}$.

- *Fusion between note and code* (Fusion): This module aggregates the representations of clinical notes and ICD codes to generate predictions, denoted as $\hat{\mathbf{y}} = \text{Fusion}(\mathbf{h}_{\text{note}}, \mathbf{h}_{\text{code}})$. To achieve this, it first applies an attention mechanism between the codes and notes by calculating $\mathbf{q}_{\text{code}} = \text{Attn}(\mathbf{h}_{\text{code}}, \mathbf{h}_{\text{note}}, \mathbf{h}_{\text{note}}) \in \mathbb{R}^{L \times d}$, where the query is the code representation $\mathbf{h}_{\text{code}}$ and the key and

value are note representation $\mathbf{h}_{\text{note}}$. It then takes the dot product between the attention output and the code representation to obtain the final output $\mathbf{o} = \mathbf{q}_{\text{code}} \odot \mathbf{h}_{\text{code}} \in \mathbb{R}^L$. Finally, a sigmoid function is applied to get the final prediction $\hat{\mathbf{y}}$.

Both $\text{Enc}_{\text{note}}$ and $\text{Enc}_{\text{code}}$ contain a text encoder $\text{Enc}_{\text{text}}$. It is a common practice to share the parameters of these two text encoders including word embeddings and model weights.

## 4 Method

We first present an algorithm to automatically extract section titles and segment clinical notes into sections. Then, we introduce the proposed training strategies for existing ICD coding models: contrastive pre-training and masked section training based on the extracted sections to reduce the variability of clinical notes with limited training data.

### 4.1 Automatic section-based segmentation

As mentioned in Section 1, clinical notes typically contain sections with standard titles, but the order of these sections may vary depending on the writing style of medical professionals. To reduce the variability in clinical notes, it is important to extract sections related to ICD codes. The initial step is to identify all possible section titles for further segmentation. However, since clinical notes are written in plain text, there are no universal rules to extract these titles. Consequently, an automatic segmentation algorithm based on the content of clinical notes is needed to extract the section titles.

Inspired by TF-IDF which can retrieve keywords in a document, we propose an n-gram document frequency-inverse average phrase frequency (DF-IAPF) algorithm to extract section titles. TF-IDF captures the unique importance of a word for a document. In TF-IDF, a word becomes a keyword of a document when it has a high term frequency in this document while few documents contain this word. However, extracting section titles is different from extracting keywords for the following reasons:

(1) Section titles are usually phrases instead of single words (e.g., "history of present illness");

(2) Unlike keywords that are common in a document but less frequent in a corpus, most clinical notes have similar section titles but they often appear only once within a clinical note.

Based on these two properties of section titles, we introduce DF-IAPF to automatically extract section titles based on the corpus-level frequency and uniqueness of phrases in the document. We first define the DF-IAPF score for a phrase $t = (w_1, w_2, \ldots, w_N)$ that contains $N$ words (n-gram).

**Document frequency-inverse average phrase frequency** We first let $\text{DF}(t)$ be the relative frequency of documents containing $t$, and $\text{IAPF}(t)$ be the inverse average phrase frequency of $t$ in all documents containing $t$:

$$\text{DF}(t) = \frac{n_t}{n_d}, \quad \text{IAPF}(t) = \frac{1}{\frac{1}{n_t}\sum_{i=1}^{n_d} f_{t,i}} = \frac{n_t}{\sum_{i=1}^{n_d} f_{t,i}}, \tag{1}$$

where $n_d$ is the total number of documents, $n_t$ is the number of documents containing $t$, and $f_{t,i}$ is the occurrence number of $t$ in the document $i$. The document frequency-inverse average phrase frequency (DF-IAPF) is defined as follows:

$$\text{DF-IAPF}(t) = \text{DF}(t) \times \text{IAPF}(t) = \frac{n_t}{n_d} \times \frac{n_t}{\sum_{i=1}^{n_d} f_{t,i}} = \frac{n_t^2}{n_d \sum_{i=1}^{n_d} f_{t,i}}. \tag{2}$$

The DF-IAPF algorithm assigns a higher score to phrases that appear frequently across all documents but occur less frequently within each document on average. For example, the formal section title, "brief hospital course", should have a higher score than a random phrase "this patient has". This is because most clinical notes contain a unique section titled "brief hospital course", while the phrase "this patient has" is more commonly used and appears multiple times in a clinical note, which lowers its score in the DF-IAPF algorithm. Then, we iterate through all n-grams with a maximum word count of $\mathcal{N}$ in clinical notes to select candidates for section titles. Since we use n-gram to extract phrases, finally, we filter out shorter titles that are subsequences of longer titles with high scores. The specific algorithm to extract candidates and complexity analysis are presented in Appendix B.

Once the section title candidates with the highest DF-IAPF scores have been retrieved, we manually select phrases from this small candidate set to form a title subset $\{t_1, t_2, \ldots, t_T\}$ with $T$ titles. This selection process is completed by medical experts to ensure the correctness of selected titles. Since section titles are mostly unique within a clinical note, we use the first occurrence position of the extracted section titles as anchors to segment each clinical note into multiple sections $\{s_k\}_{k=1}^{T}$ and build an order-agnostic structure. Given a clinical note $S$, the segmentation process from the plain text $S$ to sections $s_k$ with $n_k$ words can be summarized as follows:

$$S \xrightarrow{\text{DF-IAPF segmentation}} \{t_k : s_k\}_{k=1}^{T}, \tag{3}$$

where $t_k$ denotes a section title and $s_k = (w_1, w_2, \ldots, w_{n_k})$ is the content under the section $t_k$.

## 4.2 Supervised tree-based contrastive learning on sections

In a general ICD coding framework, `Fusion` is an attention mechanism that enables the selection of significant words in clinical notes related to code descriptions. Ideally, a clinical note should contain sections called "discharge diagnoses" and "major procedures", which encompass all code descriptions corresponding to the labels. In this case, the model can accurately extract codes from these sections. However, many clinical notes lack these two sections. Even if they exist in some clinical notes, the descriptions may be incomplete. Typically, these sections contain only primary diagnosis or procedure codes, while the labels include all secondary codes. Under these circumstances, the model must locate related records from other sections such as "physical exam" or "discharge medications", given that these sections may imply the key expressions for the diagnoses or procedures. Thus, it is necessary to improve the model's ability to comprehend the content of each section.

To accomplish this, we design a contrastive learning framework based on sections. It makes the clinical note encoder distinguish sections from the same clinical note or different clinical notes so that the model can be aware of similar clinical notes and recognize related sections.

**Construction of contrastive samples** Since we formulate ICD coding as a multi-label classification task, it is hard to find two clinical notes with the same labels as a positive pair, and it is too trivial to obtain negative pairs using two clinical notes with different labels. Therefore, we construct positive/neighbor section pairs for further training. *Positive pairs:* For each clinical note $S_i = \{t_k : s_k^i\}_{k=1}^{T}$, we randomly select an anchor section $s_k^i$. To increase the connectivity between sections in the same note, we then sample a different section $s_{k'}^i$ from the same clinical note $S_i$ to build a positive pair $(s_k^i, s_{k'}^i)$, where $t_{k'} \neq t_k$. Furthermore, from a different clinical note $S_j$, we sample two sections $s_k^j$ and $s_{k'}^j$ that correspond to $t_k$ and $t_{k'}$ in $S_i$, to build two neighbor pairs: $(s_k^i, s_k^j)$ and $(s_{k'}^i, s_{k'}^j)$. Note that, $s_k^j$ and $s_{k'}^j$ are also a positive pair. Finally, a new sample for contrastive learning is a quadruple including the anchor, positive section, and two neighbor sections, i.e., $(s_k^i, s_{k'}^i, s_k^j, s_{k'}^j)$.

**Soft multi-label similarity** To achieve the goal of contrasting section pairs in the same or different clinical notes, an intuitive idea is to calculate the Jaccard similarity between two label sets or the cosine similarity between label vectors. Unfortunately, these metrics cannot capture the underlying disease relationships in the ICD system. For example, the cosine or Jaccard similarity for two sets of disease labels: {*diabetes type I*} and {*diabetes type II*}, will be zero because they have no overlap. However, these two diseases both belong to diabetes in the ICD system. Instead of assigning hard contrastive label 0 or 1 to two label sets, we design a soft similarity of two label sets that considers disease relationships in the ICD hierarchical structure by utilizing the *tree edit distance* [31] that measures the minimum number of node edit operations (add, delete, and replace) required to transform one tree into another.

**Definition 1** (Spanning super-tree). Given the ICD hierarchical structure $\mathcal{H}$ as a tree, the label set $\mathcal{L}_i$ of clinical notes $S_i$, the spanning super-tree $\mathcal{T}_i$ of $\mathcal{L}_i$ is defined as a minimum tree that has the same root as $\mathcal{H}$ and contains all label node of $\mathcal{L}_i$ and all ancestors of $\mathcal{L}_i$: $\mathcal{T}_i = \bigcup_{l \in \mathcal{L}_i} \rho(l) \cup \mathcal{L}_i \subset \mathcal{H}$. Here, $\rho(l)$ denotes all the ancestors of one label node $l$.

Based on the spanning super-tree, the similarity $\alpha_{ij}$ between two label sets $\mathcal{L}_i$ and $\mathcal{L}_j$ is defined as:

$$\alpha_{ij} = 1 - \frac{2 \times \texttt{dist}(\mathcal{T}_i, \mathcal{T}_j)}{|\mathcal{T}_i \cup \mathcal{T}_j| - 1} \in [-1, 1], \tag{4}$$

where `dist` denotes the tree edit distance [31] between $\mathcal{T}_i$ and $\mathcal{T}_j$.

In this similarity metric, we consider both the tree edit distance and the cardinality of trees. In the denominator, we use $|\mathcal{T}_i \cup \mathcal{T}_j| - 1$ because every pair of $\mathcal{T}_i$ and $\mathcal{T}_j$ share the same root. Figure 2 shows two super-tree examples. In the first tree, nodes 5 and 7 are the labels of the clinical note $S_1$, while nodes 1, 2, and 3 are their ancestors. In the second tree, nodes 2 and 6 are labels of another clinical note, and nodes 1 and 3 are their ancestors. The tree with all colored nodes forms the spanning super-tree. The distance between these two spanning super-trees is 2 because we can delete node 5 and replace node 7 with 6 in the first tree to transform it into the second tree. Thus, the similarity of these two trees

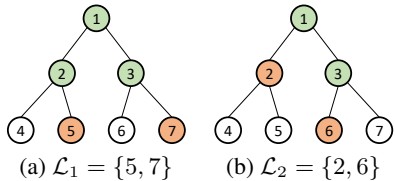

(a) $\mathcal{L}_1 = \{5, 7\}$     (b) $\mathcal{L}_2 = \{2, 6\}$

Figure 2: Examples of two spanning super-trees. Nodes with orange and green colors denote label nodes and their ancestor nodes.

is $1 - \frac{2 \times 2}{6-1} = 0.2$, since $|\mathcal{T}_i \cup \mathcal{T}_j| = |\{1, 2, 3, 5, 6, 7\}| = 6$. Although $\mathcal{L}_1$ and $\mathcal{L}_2$ do not have the same label nodes, they still share some similarities given their topological categories in the ICD hierarchical structure: the label node 2 in $\mathcal{L}_2$ is the parent of 5 in $\mathcal{L}_1$, while the label node 6 in $\mathcal{L}_2$ is a sibling of 7 in $\mathcal{L}_1$.

**Contrastive pre-training** In the pre-training of the clinical note encoder $\mathtt{Enc}_{\text{note}}$, given a quadruple $(s_k^i, s_{k'}^i, s_k^j, s_{k'}^j)$, we first calculate the note representation $\mathbf{h}_{\text{sec}}$ and apply a max pooling layer to obtain section representations for all these sections:

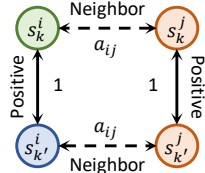

$$\mathbf{s}_{\{k,k'\}}^{\{i,j\}} = \mathtt{MaxPooling}(\mathtt{Enc}_{\text{note}}(s_{\{k,k'\}}^{\{i,j\}})) \in \mathbb{R}^d. \qquad (5)$$

Then, we aim to utilize the soft similarity between the labels of two clinical notes to guide the similarity of these four section representations. Note that the similarity between sections in a positive pair is 1 while the similarity of negative pairs is $\alpha$. Thus, we design a contrastive loss $\mathcal{J}$ using the mean absolute error (MAE) as follows:

Figure 3: Contrastive loss.

$$\mathcal{J} = \mathcal{J}_m(1, \beta(\mathbf{s}_k^i, \mathbf{s}_{k'}^i)) + \mathcal{J}_m(\alpha_{ij}, \beta(\mathbf{s}_k^i, \mathbf{s}_k^j)) + \mathcal{J}_m(1, \beta(\mathbf{s}_k^j, \mathbf{s}_{k'}^j)) + \mathcal{J}_m(\alpha_{ij}, \beta(\mathbf{s}_{k'}^i, \mathbf{s}_{k'}^j)). \quad (6)$$

Here, $\mathcal{J}_m$ denotes the MAE loss, $\alpha_{ij}$ denotes the aforementioned tree edit distance similarity between two label sets, and $\beta(\cdot, \cdot)$ is the cosine similarity function between two vectors. Figure 3 intuitively shows the proposed contrastive loss for the quadruple with soft similarity.

In summary, there are two main benefits of using contrastive learning on sections:

(1) The model gains a better understanding of the relationships between sections within a clinical note. This is particularly beneficial when a clinical note lacks explicit indicators of ICD codes, such as the "discharge diagnoses" section. With contrastive pre-training, the model can infer ICD codes by the content of related sections.

(2) It helps the model adapt to different writing styles of medical professionals and various demographic features of patients so that the model can focus on the text related to ICD codes.

### 4.3 Masked section training

As discussed earlier, existing methods mostly consider clinical notes as long sequences. Inspired by the denoising techniques for text [12], we develop a simple yet effective training strategy with permutation and masking on sections. It further mitigates the variability caused by the section order.

Given a clinical note that has been segmented into sections, denoted by $S = \{t_k : s_k\}_{k=1}^T$, we first shuffle the order of the sections to create an order-agnostic structure. Then, similar to the dropout technique [24] used to avoid overfitting in training deep learning models, we randomly mask a subset of the sections, subject to a threshold $\gamma$ where $0 \leq \gamma < 1$. The remaining sections are concatenated back into a long sequence $S'$ that is suitable for the input of existing ICD coding models. It is important to note that we do not aim to modify the original model architecture, but rather to generate samples that can help to reduce the variability of clinical notes in training. In summary, this section masking process can be described as follows:

$$S' = \bigoplus_{k \in \mathtt{perm}(T)} s_k', \quad \text{where } s_k' = \begin{cases} s_k & \text{if } |s_k| > 0 \text{ and } \theta \sim U[0,1] \geq \gamma, \\ \text{empty string} & \text{otherwise.} \end{cases} \qquad (7)$$

Table 1: Data statistics for the MIMIC-50, MIMIC-rare-50, and MIMIC-full tasks.

| Task | Item | Train | Dev | Test |
|---|---|---|---|---|
| **MIMIC-50** | # Docs. | 8,066 | 1,573 | 1,729 |
| | Avg. # words per Doc. | 1,478 | 1,739 | 1,763 |
| | Avg. # codes per Doc. | 5.7 | 5.9 | 6.0 |
| | Total # codes | 50 | 50 | 50 |
| **MIMIC-rare-50** | # Docs. | 249 | 20 | 142 |
| | Avg. # words per Doc. | 1,770 | 1,930 | 2,071 |
| | Avg. # codes per Doc. | 1.0 | 1.0 | 1.0 |
| | Total # codes | 50 | 50 | 50 |
| **MIMIC-full** | # Docs. | 47,723 | 1,631 | 3,372 |
| | Avg. # words per Doc. | 1,434 | 1,724 | 1,731 |
| | Avg. # codes per Doc. | 15.7 | 18.0 | 17.4 |
| | Total # codes | 8,692 | 3,012 | 4,085 |

Here, $\oplus$ denotes the concatenation operation, and $U[0, 1]$ means the uniform distribution from 0 to 1. By using `perm`, we can generate a random permutation of $(1, 2, \ldots, T)$ of section indices, which is a random shuffle of all sections in a clinical note.

With shuffling and masking in training, the ICD coding model is no longer limited by the order of the clinical notes. Additionally, certain sections, such as "discharge diagnoses", may not always play a deterministic role in the prediction. This allows the model to focus more on other sections that are also relevant to the predicted ICD codes. Note that, in the inference step, we do not perform shuffling and masking, but use the original sequence as input for an ICD coding model.

## 5 Experiments

### 5.1 Dataset, tasks, and evaluation metrics

The MIMIC-III [11] dataset is a popular publicly available EHR dataset that contains the discharge summaries and corresponding ground-truth ICD codes. We follow the ICD coding tasks in prior work [29, 30] and conduct three prediction tasks:

- MIMIC-50 prediction: Predicting the top 50 frequent ICD codes in the MIMIC-III dataset.

- MIMIC-rare-50 prediction: Predicting the rare 50 ICD codes that occur less than 10 times.

- MIMIC-full prediction: Predicting the entire (8,692) ICD codes in the MIMIC-III dataset.

The detailed training/dev/test dataset statistics for each task are listed in Table 1. Our experiments are conducted with cross-validation on the dev set to adjust hyper-parameters.

We use the following evaluation metrics which have been used in prior ICD coding studies [29, 30]. The metrics for MIMIC-full prediction are Macro/Micro $F_1$ and precision at 8/15 (P@8, P@15). For MIMIC-50 prediction, we use Macro/Micro $F_1$ and precision at 5 (P@5). For MIMIC-rare-50 prediction, we use Macro/Micro $F_1$.

### 5.2 Backbone models

To verify the effectiveness of our proposed section-based contrastive pre-training and masked section training (CM), we choose the following state-of-the-art ICD coding models as backbones[1]:

- **MultiResCNN** [13]: It encodes clinical notes with multi-filter residual CNN and label attention.

- **HyperCore** [4]: It also uses a convolutional encoder for text. Moreover, it applies hyperbolic embedding for ICD codes and uses GCN to model code co-occurrence.

---

[1]We do not include KEPT [29] here because our devices do not support the training of KEPT due to its high complexity. We list the result of KEPT in Appendix C.1 for reference.

Table 2: Results (%) of MIMIC-50 when trained with and without the proposed contrastive pre-training and masked training (CM) strategies. Cells with the green color denote an improvement of w/ CM compared to w/o CM.

| Model | w/o CM | | | w/ CM | | |
|---|---|---|---|---|---|---|
| | Macro $F_1$ | Micro $F_1$ | P@5 | Macro $F_1$ | Micro $F_1$ | P@5 |
| MultiResCNN | 60.8 (0.3) | 67.1 (0.1) | 64.3 (0.3) | 62.2 (0.3) | 68.1 (0.1) | 65.1 (0.2) |
| HyperCore | 61.1 (0.2) | 66.2 (0.2) | 63.5 (0.3) | 62.0 (0.2) | 67.4 (0.2) | 64.5 (0.3) |
| JointLAAT | 66.4 (0.1) | 71.6 (0.2) | 67.3 (0.4) | 67.2 (0.2) | 72.0 (0.3) | 67.9 (0.1) |
| EffectiveCAN | 66.7 (0.1) | 71.5 (0.2) | 66.4 (0.2) | 67.5 (0.2) | 71.8 (0.1) | 67.8 (0.1) |
| PLM-ICD | 64.5 (0.3) | 69.3 (0.2) | 64.5 (0.4) | 65.2 (0.1) | 70.3 (0.2) | 65.6 (0.2) |
| Hierarchical | 65.3 (0.1) | 70.6 (0.3) | 66.5 (0.1) | 66.1 (0.2) | 71.8 (0.4) | 67.2 (0.3) |
| MSMN | 68.1 (0.2) | 72.0 (0.1) | 67.5 (0.1) | 69.1 (0.1) | 72.5 (0.1) | 68.3 (0.2) |

- **JointLAAT** [25]: It uses a bidirectional LSTM to encode clinical notes and proposes a joint learning method to predict ICD codes and their parent codes in the ICD hierarchical structure.

- **EffectiveCAN** [14]: Similar to MultiResCNN, it also applies a convolutional encoder with multiple residual squeeze-and-excitation networks.

- **PLM-ICD** [9]: It is a transformer model (Roberta-base) that splits clinical notes into chunks to satisfy the maximum length of pre-trained large language models.

- **Hierarchical** [5]: It is hierarchical a transformer model (Roberta-large) by splitting clinical notes into paragraphs.

- **MSMN** [30]: It is an LSTM text encoder and incorporates the synonym of code descriptions to make the model better understand the variety of code names.

## 5.3 Implementation details

For DF-IAPF, we set the maximum word number ($\mathcal{N}$) in n-gram to 5. We set $K$ in top-$K$ candidates to 50 for the review of medical professionals. For contrastive pre-training, the batch size is 16, the learning rate is $5 \times 10^{-4}$, the optimizer is AdamW, and the epoch number is 20. The contrastive pre-training only uses the training dataset of each task to avoid data leakage. For the masked section training, we set $\gamma$ to 0.2 for MIMIC-full prediction and 0.3 for MIMIC-50/MIMIC-rare-50 prediction.

The backbone models except HyperCore [4] and EffectiveCAN [14] are implemented using their publicly released code and the optimal parameters reported in their papers. For HyperCore and EffectiveCAN, the authors do not release the code. Therefore, we implemented a version that has a close performance to the original paper. For the MIMIC-50 and MIMIC-rare-50 tasks, we run every baseline 5 times and report their average and standard deviations (std).

All programs are executed using a machine with Python 3.9.3, CUDA 11.7, an Intel i9-11900K CPU, 64GB memory, and an NVIDIA RTX 3090 GPU. The code of the proposed DF-IAPF method and training strategies can be found at: https://github.com/LuChang-CS/semi-structured-icd-coding.

## 5.4 Experimental results

**Extracted section titles** To demonstrate the effectiveness of our proposed DF-IAPF algorithm to extract section titles, we compare it with a rule-based extraction algorithm [26]. It designs special rules for every observed section title based on colons and occurrence frequencies to segment clinical notes into sections. We list the extracted section titles and analyze the effectiveness and advantages of the proposed DF-IAPF algorithm in Appendix C.2.

**MIMIC-50-prediction** We report the results of MIMIC-50 in Table 2. Here, we run each backbone model 5 times and report their mean value and std. Among all backbone models, MSMN achieves the best result on all metrics without the proposed CM. Additionally, with the proposed CM strategies, the performance of all backbone models is improved, and Macro $F_1$ is improved by 1.5% on average. Additionally, we also run a paired t-test on the Macro $F_1$ score between the backbone models w/ CM and w/o CM. The $p$-values for all backbone models are less than $5 \times 10^{-2}$, indicating that the improvement brought by the CM strategies is statistically significant over the original models.

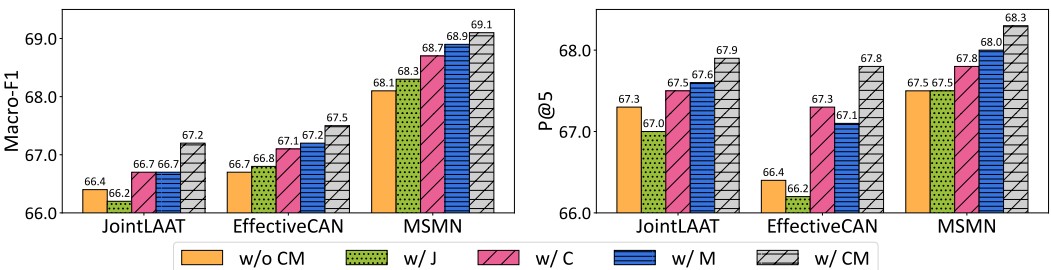

Figure 4: Abalation studies of the MIMIC-50 task when trained without or with the proposed contrastive pre-training (C) and masked section training (M). We also report a variant by replacing the tree edit distance with the simple Jaccard similarity (J).

**MIMIC-rare-50-prediction**
We report the results of MIMIC-rare-50 in Table 3. We observe that, with limited training samples in this task, the performance of backbone models is not as good as the MIMIC-50 prediction. However, the proposed CM strategies can significantly improve the prediction results. On average, the Macro $F_1$ score is improved by 55.8%. Note that, the contrastive pre-training is conducted on every task instead of the entire clinical notes. Therefore, we can conclude that the proposed CM strategies can learn a good initialization of the ICD coding models in pre-training and serve as an effective data augmentation method in training with limited data.

Table 3: Results (%) of MIMIC-rare-50 when trained with and without the proposed CM strategies. Cells with the green color denote an improvement of w/ CM compared to w/o CM.

| Model | w/o CM | | w/ CM | |
| | Macro $F_1$ | Micro $F_1$ | Macro $F_1$ | Micro $F_1$ |
|---|---|---|---|---|
| MultiResCNN | 11.2 (2.1) | 13.1 (1.8) | 22.8 (1.3) | 23.3 (1.9) |
| HyperCore | 12.5 (1.3) | 15.6 (2.2) | 23.4 (1.9) | 25.2 (1.2) |
| JointLAAT | 20.2 (1.9) | 21.6 (1.5) | 28.6 (1.1) | 27.8 (1.4) |
| EffectiveCAN | 19.8 (1.4) | 22.5 (2.1) | 27.1 (2.4) | 28.0 (1.5) |
| PLM-ICD | 22.6 (2.5) | 24.3 (1.9) | 30.3 (1.5) | 29.5 (1.3) |
| Hierarchical | 23.1 (1.7) | 24.6 (1.4) | 32.0 (1.2) | 31.3 (2.2) |
| MSMN | 23.7 (1.0) | 24.1 (1.9) | 31.2 (1.3) | 30.6 (1.7) |

Due to space constraints, we show the results of MIMIC-full in Appendix C.3. In summary, these experiments prove the capability of the DF-IAPF algorithm in extracting section titles and demonstrate the effectiveness of the proposed CM strategies in enhancing existing ICD coding models.

## 5.5 Ablation studies

To validate the effectiveness of the contrastive pre-training and masked section training, we conduct an ablation study by only applying one strategy in the MIMIC-50 prediction task and replacing the tree edit distance with Jaccard similarity. Here, we choose MIMIC-50 prediction and use JointLAAT, EffectiveCAN, and MSMN as the backbones. Figure 4 demonstrates the prediction results when training without CM (w/o CM), with only contrastive pre-training using Jaccard similarity (w/ J), with only contrastive pre-training (w/ C) using tree edit distance, with only masked section training (w/ M), and with both two strategies (w/ CM). In this figure, pre-training with simple Jaccard similarity even decreases the performance. It shows that this similarity cannot appropriately guide contrastive learning because many clinical notes have disjoint labels. We notice that both contrastive pre-training and masked section training contribute to improving the performance of the original ICD coding models. Specifically, the masked section training is slightly better than the contrastive pre-training. We think it is because the masked section training is directly applied to training ICD coding models, while the contrastive pre-training learns good initialization before training.

## 5.6 Case studies

To intuitively demonstrate the effectiveness of the proposed contrastive pre-training and masked training, in Figure 5, we give an example of a clinical note snippet and the predictions w/o CM and w/ CM using MSMN on the MIMIC-50 prediction task. Here, the ground truth contains 4 ICD

| Input | Label | w/o CM | w/ CM |
|---|---|---|---|
| Discharge Diagnosis: Esophageal cancer
...
Major Surgical or Invasive Procedure: esophagectomy
...
Social History: former smoker 40-50 ppy
...
Physical Exam: imaging:
CXR Endotracheal tube, the tip projects roughly 7 cm above the carina. | 45.13: Other endoscopy of small intestine
530.81: Esophageal reflux
96.04: Insertion of endotracheal tube
V15.82: Personal history of tobacco use | • 45.13
• 530.81 | • 45.13
• 530.81
• 96.04
• V15.82 |

Figure 5: An example of prediction without and with the proposed CM strategies using MSMN.

codes, 45.13 (procedure), 530.81 (disease), 96.04 (procedure), and V15.82 (disease). The MSMN model w/o CM predicts two codes correctly while failing to predict 96.04 and V15.82 because they do not occur in the discharge diagnosis and procedure sections. However, with CM, the MSMN model successfully predicts all four ICD codes by locating them in related sections, including "CXR Endotracheal tube" in physical exam and "former smoker" in social history.

# 6  Conclusion

In this work, we aim to minimize the variability of clinical notes in the ICD coding task by studying the semi-structured format of clinical notes. To reduce human effort, we propose an automatic algorithm to extract section titles and segment clinical notes into sections. We also design contrastive pre-training and masked section training to let the ICD coding model better locate sections related to predictions. Additionally, a tree-edit distance is designed in the loss function to measure the similarity of positive/negative pairs. Extensive experiments demonstrate the effectiveness of the proposed section title extraction algorithm and training strategies. It is worth emphasizing that our proposed methodology is versatile, as it can not only be applied to clinical notes but also employed in general multi-label classification tasks that involve semi-structures such as sections. In the future, we are committed to exploring the broader applicability of our approach across various domains.

**Limitations:** Although the proposed training strategies are able to enhance existing ICD coding models, they are dependent on the design of these models. If the model is well-designed and has many parameters, it is generally overfitting with limited training data. In this case, our proposed training strategies are a good enhancement. Additionally, we only focus on the variability caused by the order of sections in this work, but there are other formats of variability such as typos and synonyms. In the future, we plan to design new ICD coding models based on sections and consider more types of variability to further improve the robustness of the training process.

## Acknowledgments and Disclosure of Funding

This work is supported in part by the US National Science Foundation under grants 1948432, 2047843, and 2245907. Any opinions, findings, and conclusions or recommendations expressed in this material are those of the authors and do not necessarily reflect the views of the National Science Foundation.

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

# Appendix

## A    Broader Impacts

**Ethical considerations**    While EHR data contains private information of patients, the MIMIC-III dataset used in this work as well as all backbone models is a publicly available dataset. It de-identified the sensitive information of patients and doctors with masks, including admission/discharge date, name, and hospital name (e.g., [**first name3**]) to protect privacy. Therefore, the data we use will not leak such information even if we publish our code and model parameters.

**Societal Impacts**    Incorrect ICD coding can lead to medical billing errors which can affect patients and healthcare costs. However, as an enhancement of existing ICD coding models, our work aims to improve the prediction accuracy of ICD coding. We believe our method does not bring additional negative societal impacts to ICD coding.

## B    Pseudo code of the DF-IAPF algorithm

We present the proposed DF-IAPF in Algorithm 1. In lines 7-9, this algorithm uses a span in clinical notes to obtain an n-gram as a phrase $t$ and updates its occurrence in the local clinical note. Lines 10-12 update the global document frequency and phrase frequency. Finally, lines 15-16 calculate the DF-IAPF score for every phrase. In line 17 we sort all phrases descendingly by the DF-IAPF score and select the top phrases with the highest scores. Finally, we filter out shorter titles that are subsequences of longer titles with high scores in lines 18-20.

Note that this algorithm is an offline extraction of phrases before training. The computation procedures of DF-IAPF are similar to the TF-IDF, except that we add a for-loop of n-gram in line 3. Therefore, the time complexity of the DF-IAPF algorithm is $\mathcal{N} \times \mathcal{O}(\texttt{TF-IDF})$. In our experiments, the running time of the DF-IAPF algorithm is about 2 minutes.

---

**Algorithm 1:** Section Title Extraction

---

**Input**   :A set $\mathcal{S}$ of clinical notes $\mathcal{S} = \{S\}$;
         An integer $\mathcal{N}$ to control the maximum word count in n-grams;
         An integer $K$ to select top-$K$ phrases
**Output** :A candidate set $\mathcal{C}$ of section titles

1   NT $\leftarrow$ an empty mapping from phrases to counts with a default value of 0
2   APF $\leftarrow$ an empty mapping from phrases to a frequency list with a default value of an empty list
3   **for** $N \leftarrow 1$ *to* $\mathcal{N}$ **do**
4      **for** $S \in \mathcal{S}$ **do**
5          $n \leftarrow$ the number of words in $S$
6          PF $\leftarrow$ an empty mapping from phrases to frequencies with a default value of 0
7          **for** $i \leftarrow 1$ *to* $n - N + 1$ **do**
8              $t \leftarrow (w_i, w_{i+1}, \ldots, w_{i+N-1})$ // N-gram
9              $\text{PF}(t) \leftarrow \text{PF}(t) + 1$ // Update the frequency of $t$ in this document $S$
10          **for** $t \in PF$ **do**
11              $\text{NT}(t) \leftarrow \text{NT}(t) + 1$ // Update the frequency of documents containing $t$
12              Append $\text{PF}(t)$ to $\text{APF}(t)$ // Update the frequency list of $t$
13   $n_d \leftarrow |S|$
14   $\mathcal{C} \leftarrow$ an empty mapping from phrases to scores
15   **for** $t \in NT$ **do**
16      $\mathcal{C}(t) \leftarrow \frac{\text{NT}^2(t)}{n_d \times \sum_{i=1}^{\text{NT}(t)} \text{APF}(t)_i}$ // DF-IAPF, Equation (2)
17   $\mathcal{C} \leftarrow$ Sort $\mathcal{C}$ descendingly by the score and select $K$ phrases with the highest scores
18   **for** $(t_1, t_2) \in \mathcal{C} \times \mathcal{C}$ **do**
19      **if** $t_1 \subsetneq t_2$ **then**
20          $\mathcal{C} \leftarrow \mathcal{C} \setminus \{t_1\}$ // Remove shorter titles that are subsequences of longer titles with high scores.
21   **return** $\mathcal{C}$

---

Table 4: Top 20 section titles extracted by our proposed DF-IAPF algorithm and a rule-based method using colons and occurrence frequencies.

| Rank | DF-IAPF | Frequency | Rank | Rule-based | Frequency |
|---|---|---|---|---|---|
| 1 | history of present illness | 0.95 | 1 | admission date | 1.00 |
| 2 | date of birth | 0.87 | 2 | − *service* | 0.95 |
| 3 | + *sex* | 0.87 | 3 | date of birth | 0.87 |
| 4 | + *discharge date* | 1.00 | 4 | history of present illness | 0.95 |
| 5 | admission date | 1.00 | 5 | − *allergies* | 0.87 |
| 6 | social history | 0.82 | 6 | past medical history | 0.90 |
| 7 | past medical history | 0.90 | 7 | social history | 0.82 |
| 8 | discharge medications | 0.83 | 8 | − *discharge disposition* | 0.75 |
| 9 | medications on admission | 0.77 | 9 | discharge medications | 0.83 |
| 10 | discharge diagnosis | 0.94 | 10 | discharge diagnosis | 0.94 |
| 11 | discharge condition | 0.85 | 11 | medications on admission | 0.77 |
| 12 | discharge instructions | 0.71 | 12 | attending | 0.71 |
| 13 | major surgical or invasive procedure | 0.78 | 13 | family history | 0.74 |
| 14 | brief hospital course | 0.98 | 14 | discharge condition | 0.85 |
| 15 | pertinent results | 0.68 | 15 | discharge instructions | 0.71 |
| 16 | followup instructions | 0.89 | 16 | major surgical or invasive procedure | 0.78 |
| 17 | family history | 0.74 | 17 | physical exam | 0.94 |
| 18 | + *chief complaint* | 0.77 | 18 | brief hospital course | 0.98 |
| 19 | attending | 0.71 | 19 | pertinent results | 0.68 |
| 20 | physical exam | 0.94 | 20 | followup instructions | 0.89 |
| 23 | service | 0.95 | 38 | chief complaint | 0.77 |
| 28 | discharge disposition | 0.75 | 664 | discharge date | 1.00 |
| 29 | allergies | 0.87 | 1726 | sex | 1.00 |

## C   Additional experiments

### C.1   Results of KEPT

We do not include KEPT [29] in the backbone models because our devices do not support the training of KEPT due to its high complexity. We list the result of KEPT (w/o CM) here for reference. It is worth noting our proposed contrastive pre-training and masked section training are also applicable to KEPT.

- MIMIC-full prediction:
    - Macro $F_1$: 11.8
    - Micro $F_1$: 59.9
    - P@8: 77.1
    - P@15: 61.5
- MIMIC-50 prediction:
    - Macro $F_1$: 68.9
    - Micro $F_1$: 72.9
    - P@5: 67.3
- MIMIC-rare-50 prediction:
    - Macro $F_1$: 30.4
    - Micro $F_1$: 32.6

### C.2   Extracted section titles

To demonstrate the effectiveness of our proposed DF-IAPF algorithm to extract section titles, we compare it with a rule-based extraction algorithm [26]. It designs special rules for every observed section title based on colons and occurrence frequencies to segment clinical notes into sections. We list the top 20 extracted section titles in Table 4.

**Qualitative analysis**   Here, the rank is obtained using DF-IAPF scores (left) or occurrence frequencies (right). The symbol "+" indicates the title extracted by our DF-IAPF algorithm but not by the rule-based algorithm, while the symbol "−" means the title extracted by the rule-based algorithm but not the DF-IAPF algorithm in the top 20 section titles. In this Table, we observe that 17 titles are commonly extracted by both algorithms, indicating that our automatic section title algorithm is comparable to the hand-crafted rule-based method in terms of effectiveness. We further analyze the rank of missing section titles from both algorithms in the top 20 titles. All the titles that are not

Table 5: Top 20 section titles extracted by the original DF-IAPF algorithm (Raw) and titles selected by medical experts based on Raw (Selected).

| Rank | Raw | Rank | Selected |
|---|---|---|---|
| 1 | history of present illness | 1 | history of present illness |
| 2 | date of birth | 2 | date of birth |
| 3 | **sex f** | 3 | sex |
| - | **sex m** | - | |
| 4 | discharge date | 4 | discharge date |
| 5 | admission date | 5 | admission date |
| 6 | social history | 6 | social history |
| 7 | past medical history | 7 | past medical history |
| 8 | discharge medications | 8 | discharge medications |
| 9 | medications on admission | 9 | medications on admission |
| 10 | discharge diagnosis | 10 | discharge diagnosis |
| 11 | discharge condition | 11 | discharge condition |
| 12 | discharge instructions | 12 | discharge instructions |
| 13 | major surgical or invasive procedure | 13 | major surgical or invasive procedure |
| 14 | brief hospital course | 14 | brief hospital course |
| 15 | pertinent results | 15 | pertinent results |
| 16 | followup instructions | 16 | followup instructions |
| 17 | family history | 17 | family history |
| 18 | chief complaint | 18 | chief complaint |
| 19 | attending | 19 | attending |
| 20 | physical exam | 20 | physical exam |

extracted by DF-IAPF in the top 20 section titles appear in the top 30 titles. However, the titles that are missing in the rule-based method have very low ranks. It shows that even though the rules are carefully designed by humans, they may not be applicable to all clinical notes or titles. Therefore, we can conclude that our DF-IAPF algorithm is more universal than the rule-based method since it can effectively locate section titles and require less human effort.

**Quantitative analysis** To numerically demonstrate the effectiveness of our proposed DF-IAPF algorithm, we randomly select 50 clinical notes and manually extract the section title set $\Omega_i$ for each clinical note by medical experts. To evaluate the coverage of the top-20 extracted section titles $\hat{\Omega}$ by DF-IAPF and the rule-based method, we use an average intersection rate between $\Omega_i$ and $\hat{\Omega}$: $\frac{1}{50} \sum_{i=1}^{50} \frac{|\Omega_i| \cap |\hat{\Omega}|}{|\Omega|}$. The rate of DF-IAPF is 0.87, while the rate of the rule-based method is 0.83. The rates are less than 1 due to the absence of the bottom 3 titles in Table 4. Additionally, some clinical notes contain less frequent titles including "facilities", "addendum", etc. However, the rate of DF-IAPF is still higher than the rule-based method because "chief complaint", "discharge date", and "sex" are all top frequent section titles, while "discharge disposition" is a relatively less frequent title. Moreover, we report the frequency of section titles after segmentation using the 23 section titles in Table 4. We can see that all section titles have high frequencies. Together with the intersection rate, it further proves the coverage and accuracy of the extraction algorithm. Note that the rank of the section titles extracted by the rule-based method is different from the order of frequencies. This is because the rank is determined by the number of extracted section titles based on colons before segmentation. However, not all section titles are followed with a colon. Therefore, after segmentation, the frequencies may be different from title extraction.

It is worth noting that the top 20-30 titles mainly contain some special tokens, such as "[**first name3**]", which are masked tokens in the original dataset for privacy concerns. In the contrastive learning part, we do not use sections that have little relation to ICD codes, including "date of birth", "sex", "admission date", "discharge date", "attending" and "service", and use the remaining titles to pre-train the clinical note encoder. In the training of ICD coding models, we use all 23 section titles (top 20, 23, 28, and 29) so that we make the least change to the completeness of clinical notes. For some less frequent section titles such as "addendum" mentioned before, we do not segment sections by applying them as separators, but merge them with adjacent sections. In this way, the content of these sections is reserved for training.

Table 6: Results (%) of MIMIC-full when trained with/without the proposed contrastive pre-training and masked training (CM) strategies. Cells with the green color denote an improvement of w/ CM compared to w/o CM. Here, we do not provide a $p$-value since we run backbone models one time.

| Model | w/o CM | | | | w/ CM | | | |
|---|---|---|---|---|---|---|---|---|
| | Macro $F_1$ | Micro $F_1$ | P@8 | P@15 | Macro $F_1$ | Micro $F_1$ | P@8 | P@15 |
| MultiResCNN | 8.5 | 55.2 | 73.4 | 58.4 | 9.3 | 55.9 | 74.0 | 58.8 |
| HyperCore | 9.0 | 55.1 | 72.2 | 57.9 | 9.6 | 55.6 | 73.0 | 58.5 |
| JointLAAT | 10.7 | 57.5 | 73.5 | 59.0 | 11.5 | 58.3 | 73.9 | 59.4 |
| EffectiveCAN | 10.6 | 58.9 | 75.8 | 60.6 | 11.3 | 59.4 | 76.2 | 61.1 |
| PLM-ICD | 10.4 | 59.8 | 77.1 | 61.3 | 10.6 | 60.0 | 77.2 | 61.5 |
| MSMN | 10.3 | 58.4 | 75.2 | 59.9 | 11.4 | 58.8 | 75.6 | 60.2 |

**Role of medical experts discussion**    In Section 4.1, we mentioned that the selection of the extracted section title is performed by medical experts. To eliminate the selection bias, we list the originally extracted titles by our algorithm and the selected titles by medical experts in Table 5. We can see medical experts only need to correct "sex m" and "sex f". Since the extracted titles are mostly correct, there is actually little effort required by medical experts. Therefore, the role of medical experts in this process is to validate the extracted titles by the proposed DF-IAPF method, which further evaluates the effectiveness and accuracy of the DF-IAPF method.

## C.3    Results of MIMIC-full prediction

We report the results of MIMIC-full in Table 6. Here, w/o CM and w/ CM mean the results without and with the proposed CM strategies, respectively. In this task, we directly use the w/o CM results from the MSMN paper [30]. For the w/ CM results, we report the result of one run since this experiment requires a lot of time. For the results of w/o CM, all the backbone models have a relatively low Macro $F_1$ score due to the large size of the label set and long tail distribution of ICD codes, while PLM-ICD is the best in terms of Micro $F_1$, P@8, and P@15. As for the result w/ CM, the cells with green color indicate an improvement. From the comparison, we notice the proposed contrastive pre-training and masked training can improve the performance of the backbone models, among which the Macro $F_1$ score is increased by 7.1% on average. However, the PLM-ICD model does not improve as much as other backbone models. We infer it is because the PLM-ICD model already split clinical notes into chunks with a fixed length. Even with our training strategies, it somewhat breaks the information between sections so that the variability cannot be largely reduced by our proposed training strategies.

