# OpenReview forum: "Towards Semi-Structured Automatic ICD Coding via Tree-based Contrastive Learning"
_NeurIPS.cc/2023/Conference — NeurIPS 2023 poster_

### Official Review · Reviewer_BZWL · 2023-07-05

**Soundness:** 3 good
**Presentation:** 3 good
**Contribution:** 3 good
**Rating:** 6
**Confidence:** 4

**Summary:**

This paper addresses automating diagnosis coding via tree-based contrastive learning. It uses an established benchmarking dataset for this task, and achieves insightful results from its comparative performance evaluations and ablation studies.

**Strengths:**

This paper addresses automating diagnosis coding via tree-based contrastive learning. It uses an established benchmarking dataset for this task, and achieves insightful results from its comparative performance evaluations and ablation studies. It has a well chosen range of machine learning methods to compare in the coding task, and its methods are very clearly and thoroughly described.

**Weaknesses:**

The paper is somewhat lacking in its qualitative analysis. It would be helpful to extend Section 5 with a mixed method approach to analyse and evaluate the experimental methods also using qualitative approaches.

The paper is also calling for a discussion section to deepen the lessons learnt part of the study. For example, what are the limitations of the study? What are the envisioned pros and cons of the studied methods in their broader context in health and medicine? What are the ethical considerations related to using the MIMIC -III dataset?

It would have been helpful to connect this natural language processing paper to prior shared tasks and their shared datasets to allow seeing trends in methods. E.g., Computational Medicine Center's 2007 Medical NLP Challenge, followed by those by I2B2, N2C2, and CLEF eHealth would be worth of briefly surveying to assure that methods are compared to more traditional ones as well.

**Questions:**

See above

**Limitations:**

See my broader impact and ethical consideration comments above for minor comments

---

> ### Author Rebuttal · Authors · 2023-08-09
>
> ### Summary
>
> We truly appreciate your suggestions. We understand your concerns are from multiple perspectives, and we try our best to answer them in this discussion. We sincerely hope our answers can address your concerns.
>
> ---
>
> ### Weakness 1: The paper is somewhat lacking in its qualitative analysis. It would be helpful to extend Section 5 with a mixed method approach to analyse and evaluate the experimental methods also using qualitative approaches.
>
> **A:** Thank you for this suggestion. We agree that a qualitative analysis is helpful to intuitively show the effectiveness of the proposed method. In Section 5 (Experiments), we have a case study (Section 5.6) about how the proposed method can better connect labels and clinical text. Also, in Appendix B.3, Extracted section titles, we have a qualitative analysis of the extracted section titles.
>
> We hope these can be parts of the qualitative analysis. If not, we would appreciate it if the reviewer could provide more details on the specific approaches for qualitative analysis.
>
> ---
>
> ### Weakness 2 + Limitation: The paper is also calling for a discussion section to deepen the lessons learnt part of the study. For example, what are the limitations of the study? What are the envisioned pros and cons of the studied methods in their broader context in health and medicine? What are the ethical considerations related to using the MIMIC-III dataset? + See my broader impact and ethical consideration comments above for minor comments
>
> **A:** For limitations, currently we discuss them in the last paragraph of Section 6, Conclusion. But as the Reviewer ijoF suggested, we will provide a separate section to discuss more details as follows:
> > ### Limitations
> >
> > Although the proposed training strategies are able to enhance existing ICD coding models, they are dependent on the design of these models. If the model is well-designed and has many parameters, it is generally over-fitting with limited training data. In this case, our proposed training strategies are a good enhancement. Additionally, we only focus on the variability caused by the order of sections in this work, but there are other formats of variability such as typos and synonyms. In the future, we plan to design new ICD coding models based on sections and consider more types of variability to further improve the robustness of the training process.
>
> For broader impact and ethical considerations, we currently have an independent section in Appendix C, Broader Impacts. In that section, we discuss the ethical considerations of using MIMIC-III. We will consider moving them into the main paper in the future version.
> > ### Broader Impacts
> > **Ethical considerations** While EHR data contains private information of patients, the MIMIC-III dataset used in this work as well as all backbone models is a publicly available dataset. It de-identified the sensitive information of patients and doctors with masks, including admission/discharge date, name, and hospital name (e.g., [\*\*first name3\*\*]) to protect privacy. Therefore, the data we used will not leak such information even if we publish our code and model parameters.
> >
> > **Societal Impacts** Incorrect ICD coding can lead to medical billing errors which can affect patients and healthcare costs. However, as an enhancement of existing ICD coding models, our work aims to improve the prediction accuracy of ICD coding. We believe our method does not bring additional negative societal impacts to ICD coding.
>
> ---
>
> ### Weakness 3: It would have been helpful to connect this natural language processing paper to prior shared tasks and their shared datasets to allow seeing trends in methods. E.g., Computational Medicine Center's 2007 Medical NLP Challenge, followed by those by I2B2, N2C2, and CLEF eHealth would be worth of briefly surveying to assure that methods are compared to more traditional ones as well.
>
> **A:** Thank you for this suggestion. We understand it will be good to incorporate traditional datasets and models. Currently, we focus on following the routine of the recently published backbone models used in this paper. Since they mainly use the MIMIC dataset, we think it is fair to compare the performance on the MIMIC dataset in this work. But we will definitely explore the I2B2, N2C2, and CLEF eHealth datasets in future work.

---

> > ### Comment · Reviewer_BZWL · 2023-08-15
> > **Rebuttal response**
> >
> > Based on the clear and convincing response by the authors, I have revised my review scoring.

---

> > > ### Author Response · Authors · 2023-08-15
> > >
> > > We sincerely appreciate your recognition and valuable suggestions in this review!

---

### Official Review · Reviewer_cSeG · 2023-07-06

**Soundness:** 3 good
**Presentation:** 3 good
**Contribution:** 3 good
**Rating:** 6
**Confidence:** 3

**Summary:**

This paper describes a novel method of ICD coding that explicitly model clinical note sections. Instead of treating a clinical note as a long sequence of tokens, the authors propose to segment a clinical note into sections and then use contrasive learning to pre-train the model.
Experiment results on MIMIC-III show that the proposed components can be used to improve the effectiveness of several existing CNN, RNN, Transformer-based models, especially when the training data is limited.

**Strengths:**

* A simple yet effective contrastive learning variant based on label tree
* The proposed (section segmentation and contrastive learning) components are used with several existing models and are shown to be effective, especially with limited training data.

**Weaknesses:**

* The used baselines are mainly CNN, RNN based. The only used transformer-based baseline PLM-ICD seems not to be a strong one. See suggestion 3

**Questions:**

* Suggest swapping equations (1) and (2)
* Section 4.3: can you explain what is the role of perm operation (or saying, why it is necessary)?
* Suggest to consider stronger transformer-based baselines [1]

[1]     Xiang Dai, Ilias Chalkidis, Sune Darkner, Desmond Elliott, "Revisiting Transformer-based Models for Long Document Classification", in Findings of EMNLP, 2022.

---

> ### Author Rebuttal · Authors · 2023-08-09
>
> ### Summary
>
> We are glad to know you think our work is effective. We truly appreciate your suggestions. We believe your concerns are mainly due to the baseline selection, especially for Transformer-based models. We carefully read your suggested paper and add the result of a stronger Transformer-based model. We sincerely hope it can adequately address your concerns.
>
> ---
>
> ### Weakness + Question 3: The used baselines are mainly CNN, RNN based. The only used transformer-based baseline PLM-ICD seems not to be a strong one. See suggestion 3 + Suggest to consider stronger transformer-based baselines:
> [1] Xiang Dai, Ilias Chalkidis, Sune Darkner, Desmond Elliott, "Revisiting Transformer-based Models for Long Document Classification", in Findings of EMNLP, 2022.
>
> **A:** Thank you for these suggestions. We first list Table 2 in that paper for your reference (C: CNN, T: Transformer, R: RNN, L: Roberta-large). It is worth noting that the RNN-based MSMN model (one of the baselines used in our paper) has the best performance even compared to various Transformer-based models. This indicates that we already use strong baselines.
>
> | Model          | Model Type  | Macro AUC | Micro AUC | Macro F1 | Micro F1 | P@5  |
> |----------------|-------------|-----------|-----------|----------|----------|------|
> | CAML           |  C          | 88.4      | 91.6      | 57.6     | 63.3     | 61.8 |
> | PubMedBERT     |  T          | 88.6      | 90.8      | 63.3     | 68.1     | 64.4 |
> | GatedCNN-NCI   |  C          | 91.5      | 93.8      | 62.9     | 68.6     | 65.3 |
> | LAAT           |  R          | 92.5      | 94.6      | 66.6     | 71.5     | 67.5 |
> | **MSMN**           |  R          | **92.8**      | **94.7**      | **68.3**     | **72.5**     | **68.0** |
> | Baselines processing up to 512 tokens    |           |          |          |      |
> | First          |  T          | 83.0      | 86.0      | 47.0     | 56.1     | 55.4 |
> | Random         |  T          | 82.5      | 85.4      | 42.7     | 51.1     | 52.3 |
> | Informative    |  T          | 82.7      | 85.8      | 46.4     | 55.2     | 54.8 |
> | Long document models         |           |           |          |          |      |
> | Longformer (4096 + LWAN)     |  T          | 90.0      | 92.6      | 60.7     | 68.2     | 64.8 |
> | Hierarchical (4096 + LWAN)    |  T          | 91.1      | 93.6      | 62.9     | 69.5     | 65.7 |
> | Hierarchical (4096 + LWAN + L)   |  T          | 91.7      | 94.1      | 65.2     | 71.0     | 66.2 |
> | Hierarchical (4096 + LWAN)   |  T          | 91.4      | 93.7      | 63.8     | 70.1     | 65.9 |
> | Hierarchical (4096 + LWAN + L)   |  T          | 91.9      | 94.1      | 65.5     | 71.1     | 66.4 |
>
> We choose PLM-ICD because it is a recently published Transformer-based model (2022), which shows strong performance in the MIMIC-full setting. PLM-ICD splits the clinical notes into chunks. It can also be considered as a type of Hierarchical Transformer. However, since it uses Roberta-base instead of Roberta-large (L), the performance is similar to Hierarchical (4096 + LWAN) in this table.
>
> To make a stronger comparison, besides PLM-ICD, we have added Hierarchical (4096 + LWAN + L) as another backbone model. Here, LWAN refers to “label-wise attention network”. We list the results as follows:
>
> For MIMIC-50:
>
> | Model        | w/o CM      |             |            | w/ CM      |             |            |           |
> |--------------|-------------|-------------|------------|-------------|-------------|------------|-----------|
> |              | Macro-$F_1$ | Macro-$F_1$ | P@5        | Macro-$F_1$ | Macro-$F_1$ | P@5        | $p$-value |
> | PLM-ICD      | 64.5 (0.3)  | 69.3 (0.2)  | 64.5 (0.4) | 65.2 (0.1) | 70.3 (0.2) | 65.6 (0.2) | $2 \times 10^{-4}$  |
> | Hierarchical  | 65.3 (0.1)  | 70.6 (0.3)  | 66.5 (0.1) | 66.1 (0.2) | 71.8 (0.4) | 67.2 (0.3) | $4 \times 10^{-4}$ |
>
> For MIMIC-rare-50:
>
> | Model        | w/o CM      |             | w/ CM      |             |           |
> |--------------|-------------|-------------|-------------|-------------|-----------|
> |              | Macro-$F_1$ | Macro-$F_1$ | Macro-$F_1$ | Macro-$F_1$ | $p$-value |
> | PLM-ICD      | 22.6 (2.5)  | 24.3 (1.9)  | 30.3 (1.5)  | 29.5 (1.3)  | $6 \times 10^{-4}$  |
> | Hierarchical  | 23.1 (1.7)  | 24.6 (1.4)  | 32.0 (1.2)  | 31.3 (2.2)  | $8 \times 10^{-5}$  |
>
> In the table for MIMIC-50, the hierarchical results are slightly different from Table 2 above. We think it is because of different random initialization of different runs. Nevertheless, for the hierarchical transformer, we can still observe significant improvement with small $p$-values.
>
> ---
>
> ### Question 1: Suggest swapping equations (1) and (2)
>
> **A:** Thank you for this suggestion. We will swap equations (1) and (2) and update expressions for better clarity.
>
> ---
>
> ### Question 2: Section 4.3: can you explain what is the role of perm operation (or saying, why it is necessary)?
>
> **A:** We apologize for any confusion here. By using `perm` to get a random permutation of section indices, we want to generate a random shuffle of all sections in a clinical note. This shuffling is used as a denoising technique in the BART paper. We will clarify this in the future version.

---

> > ### Comment · Reviewer_cSeG · 2023-08-13
> >
> > Thanks for providing additional results. I do not have other major concerns.

---

> > > ### Author Response · Authors · 2023-08-15
> > >
> > > We sincerely thank your effort and valuable suggestions in this review!

---

### Official Review · Reviewer_VQpM · 2023-07-07

**Soundness:** 2 fair
**Presentation:** 2 fair
**Contribution:** 2 fair
**Rating:** 3
**Confidence:** 4

**Summary:**

The paper proposed a semi-structured automatic ICD coding algorithm with a contrastive pre-training and masked section training and evaluate the algorithm using MIMIC-III dataset.

**Strengths:**

The paper is well structured with clear explanation in research motivation, related work, experiment configuration and results.
Empirical results were obtained with multiple baseline models on benchmark dataset MIMIC-III.
Code and data are also provided.

**Weaknesses:**

1. In the related work part, the paper misses the weak supervision approach applied on ICD coding.
2. Result tables (Table 1 and Table 2) and result plots (Figure 4) miss confidence intervals (CIs). Standard deviations are not intuitive to illustrate the variance for model comparison, especially only 5 repetitions are performed. Pls compute the CIs for each results, which should be straight forward.
3. The comparison of w/C and w/M in Figure 4 is interesting to illustrate the utility and necessity of section identification in your proposed framework. Better to provide a table with results and confidence intervals in main text or supplement. Otherwise, the performance difference looks very trivial.
In general, the model evaluation is weak in the current representation, which might eliminate the technical soundness.


**Questions:**

1. Are the results cross validated? It is not specified in the main text or supplement. Pls justify the significance and generalizability of the proposed model.

**Limitations:**

Yes. Limitations are addressed in the paper.

---

> ### Author Rebuttal · Authors · 2023-08-09
>
> ### Summary
>
> We truly appreciate your suggestions and understand your concerns mainly come from the result presentation. We have added the confidence intervals for Table 1 and Table 2. We have also added one representative table with confidence intervals of Figure 4. We sincerely hope these updates can adequately address your concerns.
>
> ---
>
> ### Weakness 1: In the related work part, the paper misses the weak supervision approach applied on ICD coding.
>
> **A:** Thank you for this suggestion. We will add the following papers and corresponding discussions to the related work part:
> - Dong, Hang, et al. "Rare disease identification from clinical notes with ontologies and weak supervision." 2021 43rd Annual International Conference of the IEEE Engineering in Medicine & Biology Society (EMBC). IEEE, 2021.
> - Cusick, Marika, et al. "Using weak supervision and deep learning to classify clinical notes for identification of current suicidal ideation." Journal of psychiatric research 136 (2021): 95-102.
> - Gao, Chufan, et al. "Classifying unstructured clinical notes via automatic weak supervision." Machine Learning for Healthcare Conference. PMLR, 2022.
>
> ---
>
> ### Weakness 2: Result tables (Table 1 and Table 2) and result plots (Figure 4) miss confidence intervals (CIs). Standard deviations are not intuitive to illustrate the variance for model comparison, especially only 5 repetitions are performed. Pls compute the CIs for each results, which should be straight forward.
>
> **A:** We appreciate this suggestion. In the following tables, we have added the confidence interval (95%) of the paired *t*-test for Table 1 and Table 2. Here, the confidence interval for the paired *t*-test denotes we have 95% confidence that the difference of the average Macro-$F_1$  score is in this interval. We believe that, together with $p$-value, the value of confidence intervals can be evidence that our method is a significant improvement over the backbone models.
>
> Confidence Interval for Table 1 (MIMIC-50):
>
> | Model | Macro-$F_1$ w/o CM | Macro-$F_1$ w/ CM | $p$-value | Confidence Interval |
> |--------------|--------------------|-------------------|--------------------|---------------------|
> | MultiResCNN | 60.8 (0.3) | 62.2 (0.3) | $1 \times 10^{-4}$ | [1.2, 2.3] |
> | HyperCore | 61.1 (0.2) | 62.0 (0.2) | $5 \times 10^{-3}$ | [0.4, 1.4] |
> | JointLAAT | 66.4 (0.1) | 67.2 (0.2) | $7 \times 10^{-3}$ | [0.4, 1.3] |
> | EffectiveCAN | 66.7 (0.1) | 67.5 (0.2) | $3 \times 10^{-3}$ | [0.4, 0.9] |
> | PLM-ICD | 64.5 (0.3) | 65.2 (0.1) | $2 \times 10^{-4}$ | [0.4, 0.7] |
> | MSMN | 68.1 (0.2) | 69.1 (0.1) | $8 \times 10^{-4}$ | [0.7, 1.3] |
>
> Confidence Interval for Table 2 (MIMIC-rare-50):
>
> | Model | Macro-$F_1$ w/o CM | Macro-$F_1$ w/ CM | $p$-value | Confidence Interval |
> |--------------|--------------------|-------------------|--------------------|---------------------|
> | MultiResCNN | 11.2 (2.1) | 22.8 (1.3) | $5 \times 10^{-4}$ | [9.4, 16.0] |
> | HyperCore | 12.5 (1.3) | 23.4 (1.9) | $3 \times 10^{-5}$ | [10.6, 13.7] |
> | JointLAAT | 20.2 (1.9) | 28.6 (1.1) | $2 \times 10^{-4}$ | [8.3, 13.3] |
> | EffectiveCAN | 19.8 (1.4) | 27.1 (2.4) | $1 \times 10^{-4}$ | [6.9, 10.7] |
> | PLM-ICD | 22.6 (2.5) | 30.3 (1.5) | $6 \times 10^{-4}$ | [6.3, 11.0] |
> | MSMN | 23.7 (1.0) | 31.2 (1.3) | $2 \times 10^{-5}$ | [8.7, 10.8] |
>
> ---
>
> ### Weakness 3: The comparison of w/C and w/M in Figure 4 is interesting to illustrate the utility and necessity of section identification in your proposed framework. Better to provide a table with results and confidence intervals in main text or supplement. Otherwise, the performance difference looks very trivial. In general, the model evaluation is weak in the current representation, which might eliminate the technical soundness.
>
> **A:** We completely understand the importance of confidence intervals. Here, we have added confidence interval for the variants of MSMN in terms of Macro-$F_1$  score. Please understand it will be a large table if we list all variants of all models. Therefore, due to the space limit, we choose the confidence interval of MSMN variants as a representative. We will add the full table to the main paper and supplementary in the future version.
>
> | Variant | Macro-$F_1$ of Variant | Confidence Interval for Macro-$F_1$ w/CM: 69.1 (0.1) |
> |---------|------------------------|------------------------------|
> | w/o CM | 68.1 (0.2) | [0.7, 1.3] |
> | w/ J | 68.3 (0.1) | [0.6, 1.0] |
> | w/ C | 68.7 (0.2) | [0.3, 0.8] |
> | w/ M | 68.9 (0.1) | [0.2, 0.4] |
>
> ---
>
> ### Question: Are the results cross validated? It is not specified in the main text or supplement. Pls justify the significance and generalizability of the proposed model.
>
> **A:** In the Appendix, Dataset Statistics (Table 3), we demonstrated that the dataset is split into training, dev (validation), and test data. We follow the dataset split settings in CAML [1] and KEPT [2], which are random split. Our experiments are conducted with cross-validation on the dev set to adjust hyper-parameters. We will clarify this in the main paper.
>
> It is worth noting that we strictly follow the basic rules for training deep learning models in experiments. These rules include but are not limited to random dataset split, cross-validation, multiple runs with different random seeds, and comparing with strong baselines. We believe together with the suggested confidence intervals, these rules can ensure the significance and generalizability of the proposed model.
>
> > [1] Mullenbach, James, et al. "Explainable Prediction of Medical Codes from Clinical Text." Proceedings of NAACL 2018.
> >
> > [2] Yang, Zhichao, et al. "Knowledge Injected Prompt Based Fine-tuning for Multi-label Few-shot ICD Coding." arXiv preprint arXiv:2210.03304 (2022).

---

> > ### Author Response · Authors · 2023-08-18
> >
> > Dear Reviewer VQpM,
> >
> > We would like to thank you again for your reviews. We understand reviewing is a time-consuming process. Your feedback on our rebuttal is more than valuable in improving the quality of our paper. If there are any further concerns or questions, please feel free to let us know before the author discussion period ends. We will be happy to answer them during the discussion.
> >
> > Thank you!

---

> > ### Comment · Reviewer_VQpM · 2023-08-21
> > **Thank you**
> >
> > Thank you for providing the answers to the questions as well as confidence intervals for the experiments. However, it seems that a two-side t-test is performed to check the difference in distributions, instead of a one-side t-test to prove that the performance is improved from one to the other. It's not enough to indicate the proposed method is better than baseline. Pls justify.  Thank you!

---

> > > ### Author Response · Authors · 2023-08-21
> > >
> > > To justify, we further calculate one-sided (greater) paired *t*-test using scipy's `ttest_rel` function:
> > > ```python
> > > result = ttest_rel(a, b, alternative='greater')
> > > print('p-value:', result.pvalue)
> > > print('confidence interval:', result.confidence_interval(confidence_level=0.95))
> > > ```
> > > Here, `a` is the results of our method, and `b` is the results of the backbone methods. We demonstrate the new p-values and confidence intervals as follows:
> > >
> > > Confidence Interval for Table 1:
> > >
> > > | Model        | Macro-$F_1$ w/o CM | Macro-$F_1$ w/ CM | $p$-value          | Confidence Interval |
> > > |--------------|--------------------|-------------------|--------------------|---------------------|
> > > | MultiResCNN  | 60.8 (0.3)         | 62.2 (0.3)        | $6 \times 10^{-5}$ | [1.3, $+\infty$]          |
> > > | HyperCore    | 61.1 (0.2)         | 62.0 (0.2)        | $3 \times 10^{-3}$ | [0.5, $+\infty$]          |
> > > | JointLAAT    | 66.4 (0.1)         | 67.2 (0.2)        | $4 \times 10^{-3}$ | [0.5, $+\infty$]          |
> > > | EffectiveCAN | 66.7 (0.1)         | 67.5 (0.2)        | $1 \times 10^{-3}$ | [0.5, $+\infty$]          |
> > > | PLM-ICD      | 64.5 (0.3)         | 65.2 (0.1)        | $1 \times 10^{-4}$ | [0.4, $+\infty$]          |
> > > | MSMN         | 68.1 (0.2)         | 69.1 (0.1)        | $4 \times 10^{-4}$ | [0.9, $+\infty$]          |
> > >
> > > Confidence Interval for Table 2:
> > >
> > > | Model        | Macro-$F_1$ w/o CM | Macro-$F_1$ w/ CM | $p$-value          | Confidence Interval |
> > > |--------------|--------------------|-------------------|--------------------|---------------------|
> > > | MultiResCNN  | 11.2 (2.1)         | 22.8 (1.3)        | $2 \times 10^{-4}$ | [9.8, $+\infty$]         |
> > > | HyperCore    | 12.5 (1.3)         | 23.4 (1.9)        | $1 \times 10^{-5}$ | [11.6, $+\infty$]        |
> > > | JointLAAT    | 20.2 (1.9)         | 28.6 (1.1)        | $9 \times 10^{-5}$ | [8.8, $+\infty$]         |
> > > | EffectiveCAN | 19.8 (1.4)         | 27.1 (2.4)        | $6 \times 10^{-5}$ | [7.2, $+\infty$]         |
> > > | PLM-ICD      | 22.6 (2.5)         | 30.3 (1.5)        | $3 \times 10^{-4}$ | [6.8, $+\infty$]         |
> > > | MSMN         | 23.7 (1.0)         | 31.2 (1.3)        | $1 \times 10^{-5}$ | [9.2, $+\infty$]         |
> > >
> > > Confidence Interval for Figure 4 (MSMN):
> > >
> > > | Variant | Macro-$F_1$ of Variant | Confidence Interval for Macro-$F_1$ w/CM: 69.1 (0.1) |
> > > |---------|------------------------|------------------------------|
> > > | w/o CM  | 68.1 (0.2)             | [0.8, $+\infty$]                   |
> > > | w/ J    | 68.3 (0.1)             | [0.6, $+\infty$]                   |
> > > | w/ C    | 68.7 (0.2)             | [0.4, $+\infty$]                   |
> > > | w/ M    | 68.9 (0.1)             | [0.2, $+\infty$]                   |
> > >
> > > We hope the new results can adequately address your concerns.

---

### Official Review · Reviewer_ijoF · 2023-07-26

**Soundness:** 4 excellent
**Presentation:** 4 excellent
**Contribution:** 3 good
**Rating:** 7
**Confidence:** 4

**Summary:**

The paper tackles automatic ICD coding. It lists challenges and proposes solutions to them:
1. For ignoring structural information, it proposes a content-based algorithm that automatically segments clinical notes into sections.
2. For limited availability of data and variability of clinical notes, it proposes a contrastive learning framework based on a soft multi-label similarity with tree edit distance and a masked section training strategy.

The authors conduct extensive experiments on MIMIC-III and demonstrate that their proposed methods can enhance the performance of existing ICD coding models.

**Strengths:**

- The paper is well-written and easy to follow. The introduction, related work and preliminaries gives a clear background and motivation to the problem.
- Code is provided in the supplementary material.
- DF-IAPF is neat, easy to implement, and fast to run.
- DF-IAPF has two assumptions: 1) section titles have high document frequency; 2) section titles have low average phrase frequency. While there are corner uses where a section title only appears in some notes, or it appears multiple times in a note, the paper provides both qualitative and quantitive analysis that compare DF-IAPT with a rule-based algorithm and demonstrates the effectiveness of DF-IAPT.
- Soft multi-label similarity based on tree edit distance captures the hierarchy difference between two ICD label sets.
- The choices of tasks cover frequent, rare, and the entire ICD codes in MIMIC-III.
- Experiments verify the effectiveness of proposed methods on a diverse set of backbones.
- Ablation studies clearly show the improvement of each method proposed in this paper.

**Weaknesses:**

An expert selection process is needed after DF-IAPF proposes a candidate set. How important is the expert selection process? There should be a comparison between DF-IAPF fully automatic and DF-IAPF followed by an expert selection.

**Questions:**

- Line 238: Is h_section more accurate than h_note?
- Typo in line 12 of Algorithm 1 in Appendix A: TF(t) -> PF(t)?

**Limitations:**

This paper includes a section in Appendix, discussing broader impacts. The conclusion section also briefly states the limitations of this work. It would be better to have a separate section to discuss these limitations.

---

> ### Author Rebuttal · Authors · 2023-08-09
>
> ### Summary
>
> We are delighted to know that you think our work has multiple strengths. We understand that your primary concern is the role of medical experts in the title selection. To address this, we have added the comparison between the titles extracted by our algorithm and those selected by medical experts. The results demonstrate that the titles derived from our algorithm align closely with the titles chosen by experts. This evidence confirms that the selection process demands minimal effort from experts. We sincerely hope these results can adequately address your concerns.
>
> ---
>
> ### Weaknesses: An expert selection process is needed after DF-IAPF proposes a candidate set. How important is the expert selection process? There should be a comparison between DF-IAPF fully automatic and DF-IAPF followed by an expert selection.
>
> **A:** We agree it is important to discuss the role of medical experts in candidate selection. Thank you for your suggestion. We list the originally extracted titles by our algorithm and the selected titles by medical experts as follows:
>
> | Rank | Original Title                       | Rank | Selected Title By Medical Experts    |
> |------|------------------------------------- |------|--------------------------------------|
> |  1   | history of present illness           |  1   | history of present illness           |
> |  **2**   | **sex f**                        |  2   | sex                                  |
> |  **3**   | **sex m**                        |  -   |                                      |
> |  4   | date of birth                        |  3   | date of birth                        |
> |  5   | discharge date                       |  4   | discharge date                       |
> |  6   | admission date                       |  5   | admission date                       |
> |  7   | social history                       |  6   | social history                       |
> |  8   | past medical history                 |  7   | past medical history                 |
> |  9   | discharge medications                |  8   | discharge medications                |
> |  10  | medications on admission             |  9   | medications on admission             |
> |  11  | discharge diagnosis                  |  10  | discharge diagnosis                  |
> |  12  | discharge condition                  |  11  | discharge condition                  |
> |  13  | discharge instructions               |  12  | discharge instructions               |
> |  14  | major surgical or invasive procedure |  13  | major surgical or invasive procedure |
> |  15  | brief hospital course                |  14  | brief hospital course                |
> |  16  | pertinent results                    |  15  | pertinent results                    |
> |  17  | followup instructions                |  16  | followup instructions                |
> |  18  | family history                       |  17  | family history                       |
> |  19  | chief complaint                      |  18  | chief complaint                      |
> |  20  | attending                            |  19  | attending                            |
> |  21  | physical exam                        |  20  | physical exam                        |
>
> We can see medical experts only need to correct  `sex m`  and `sex f`. Since the extracted titles are mostly correct, there is actually little effort required by medical experts. Therefore, the role of medical experts in this process is to validate the extracted titles by the proposed DF-IAPF method, which further evaluates the effectiveness and accuracy of the DF-IAPF method.
>
> To clarify, we will also add this table to Appendix in the future version.
>
> ---
>
> ### Question 1: Line 238: Is h_section more accurate than h_note?
>
> **A:** Thank you for this suggestion! We used $h_{note}$  as a general symbol for output of $Enc_{note}$ . But here, we agree that $h_{sec}$  is more accurate. We will update this notation in the future version.
>
> ---
>
> ### Question 2: Typo in line 12 of Algorithm 1 in Appendix A: TF(t) -> PF(t)?
>
> **A:** Thank you for this correction. We will fix this in the future version.
>
> ---
>
> ### Limitations: This paper includes a section in Appendix, discussing broader impacts. The conclusion section also briefly states the limitations of this work. It would be better to have a separate section to discuss these limitations.
>
> **A:** Thank you for this suggestion! We agree that it will be better to add an independent section about limitations. We will add it in the future version, and it will look like this:
> > ### Limitations
> >
> > Although the proposed training strategies are able to enhance existing ICD coding models, they are dependent on the design of these models. If the model is well-designed and has many parameters, it is generally over-fitting with limited training data. In this case, our proposed training strategies are a good enhancement. Additionally, we only focus on the variability caused by the order of sections in this work, but there are other formats of variability such as typos and synonyms. In the future, we plan to design new ICD coding models based on sections and consider more types of variability to further improve the robustness of the training process.

---

> > ### Author Response · Authors · 2023-08-18
> >
> > Dear Reviewer ijoF,
> >
> > We would like to thank you again for your feedback. If there are any further concerns or questions, please do not hesitate to let us know before the author discussion period ends. We will be happy to answer them during the discussion.
> >
> > Thank you!

---

> > ### Comment · Reviewer_ijoF · 2023-08-19
> >
> > Thanks for your responses. I raised my score to 7. I do not have other major concerns.

---

> > > ### Author Response · Authors · 2023-08-19
> > >
> > > We sincerely thank your great effort and valuable suggestions in this review!

---

### Decision · Program_Chairs · 2023-09-21

**Decision:**

Accept (poster)

**Comment:**

Thank you for your submission.

Reviewers noted a number of strengths including clarity of writing (ijoF, VQpM), simplicity of the approach (ijoF, cSeG), appropriate evaluation (ijoF), and useful ablations (ijoF). Reviewers raised a concern around a weak transformer baseline (cSeG), which was addressed well in the rebuttal. Concerns were raised regarding omitting certain areas such as learning with noisy labels (VQpM) and qualitative analysis (BZWL). A lack of discussion around potential limitations and impact of the approach was also highlighted (ijoF, BZWL), also well addressed by the authors. Statistical quantification was discussed as well (VQpM).

Overall this was a well received paper where reviewer concerns were well addressed, and I recommend acceptance.

P.S. There was a lengthy discussion around statistically comparing classifiers (VQpM). I would actually caution both the reviewer and the authors around blindly applying statistical tests, and to think deeply about appropriate statistical comparisons of machine learning models.
Recommended reading: https://www.jmlr.org/papers/volume7/demsar06a/demsar06a.pdf